# Single-Cell Next-Generation Sequencing to Monitor Hematopoietic Stem-Cell Transplantation: Current Applications and Future Perspectives

**DOI:** 10.3390/cancers15092477

**Published:** 2023-04-26

**Authors:** Olisaemeka Ogbue, Serhan Unlu, Gogo-Ogute Ibodeng, Abhay Singh, Arda Durmaz, Valeria Visconte, John C. Molina

**Affiliations:** 1Internal Medicine, Cleveland Clinic Fairview Hospital, Cleveland, OH 44111, USA; ogbueo@ccf.org (O.O.); unlus@ccf.org (S.U.); 2Internal Medicine, Infirmary Health’s Thomas Hospital, Fairhope, AL 36607, USA; gogo.ibodeng@infirmaryhealth.org; 3Department of Hematology Medical Oncology, Taussig Cancer Center, Cleveland Clinic, Cleveland, OH 44106, USA; singha21@ccf.org (A.S.); molinaj3@ccf.org (J.C.M.); 4Department of Translational Hematology and Oncology Research, Cleveland Clinic Taussig Cancer Center, Cleveland, OH 44106, USA

**Keywords:** acute myeloid leukemia, myelodysplastic syndrome/neoplasm, hematopoietic stem-cell transplantation, measurable residual disease

## Abstract

**Simple Summary:**

Single-cell DNA sequencing is a laboratory technique that analyzes the genetic content of individual cells. In the context of genetically diverse hematological cancers such as acute myeloid leukemia (AML) and myelodysplastic syndrome (MDS), the traditional approach of analyzing genetic material, which typically involves bulk samples of leukemia cells, may miss important mutations that may not be present in all cells. The single-cell DNA technique can better identify genetic mutations associated with disease recurrence and death even after stem-cell transplantation. The paper discusses the background, pitfalls, and applications of this technology when used during transplantation procedures. The use of this technology could potentially translate to better outcomes in AML/MDS patients receiving stem-cell transplantation.

**Abstract:**

Acute myeloid leukemia (AML) and myelodysplastic syndrome (MDS) are genetically complex and diverse diseases. Such complexity makes challenging the monitoring of response to treatment. Measurable residual disease (MRD) assessment is a powerful tool for monitoring response and guiding therapeutic interventions. This is accomplished through targeted next-generation sequencing (NGS), as well as polymerase chain reaction and multiparameter flow cytometry, to detect genomic aberrations at a previously challenging leukemic cell concentration. A major shortcoming of NGS techniques is the inability to discriminate nonleukemic clonal hematopoiesis. In addition, risk assessment and prognostication become more complicated after hematopoietic stem-cell transplantation (HSCT) due to genotypic drift. To address this, newer sequencing techniques have been developed, leading to more prospective and randomized clinical trials aiming to demonstrate the prognostic utility of single-cell next-generation sequencing in predicting patient outcomes following HSCT. This review discusses the use of single-cell DNA genomics in MRD assessment for AML/MDS, with an emphasis on the HSCT time period, including the challenges with current technologies. We also touch on the potential benefits of single-cell RNA sequencing and analysis of accessible chromatin, which generate high-dimensional data at the cellular resolution for investigational purposes, but not currently used in the clinical setting.

## 1. Introduction

Acute myeloid leukemia (AML) and myelodysplastic syndrome (MDS) are heterogeneous clinical and molecular entities that share many recurrent driver mutations [1,2,3]. The clonal architecture of these diseases creates a challenge in monitoring treatment response, particularly in the presence of measurable residual disease (MRD), which refers to the persistence of leukemic cells in the bone marrow or peripheral blood of patients in morphologic complete remission after treatment [4].

Allogeneic hematopoietic stem-cell transplant (AlloHSCT) is a potentially curative treatment for adults with high-risk/refractory AML/MDS, but disease relapse is a significant cause of transplant failure. In AML, incidence of relapse approaches 25–55% [5], with a 2 year overall survival of 14–25% in relapsed patients [6]. In MDS, a 3 year cumulative incidence of relapse (CIR) of 37% was observed in patients receiving either induction chemotherapy or demethylating agents (DMAs) before HSCT [7].

The detection of MRD in both conditions also correlates with survival and risk of relapse according to a growing body of evidence [8,9]. In the pre-transplantation timeframe, MRD status can guide therapeutic interventions, such as the intensity of pre-transplant conditioning or post-transplant chemotherapy/immunosuppression [10,11]. MRD monitoring is now recommended as routine follow-up post allogeneic HSCT to identify candidate subjects who would benefit from further treatment [12].

The prognostic significance of MRD detection is further reinforced by its inclusion in risk-stratifying patients according to European LeukemiaNet (ELN) consensus guidelines for AML diagnosis and management. In the updated 2022 ELN guidelines, persistence of MRD after complete treatment response (CRMRD+) raises the ELN risk classification [13].

Single-cell sequencing (scSeq) can reveal the clonal and subclonal architecture of diseases that are difficult to study using bulk sequencing methods. The clinical utility of scSeq in monitoring MRD has only recently been tested in AML/MDS, with promising results in contemporary studies [14,15]. However, clinical application of scSeq is yet to be standardized due to limitations with current technologies.

This review discusses the latest updates on the application of single-cell DNA genomics in monitoring MRD in AML and MDS. The emphasis is on the HSCT period, including challenges with current technologies and future perspectives on exploratory technologies such as single-cell RNA sequencing (scRNA-seq) and single-cell sequencing of chromatin-accessible regions (scATAC-seq).

## 2. Genomic/Immunophenotypic Alterations during HSCT and Mechanisms of Relapse

Significant changes occur in the genomic and immunophenotypic landscape of AML/MDS during allogeneic HSCT. Transplantation involves combination chemotherapy (conditioning) to eliminate normal and leukemia stem cells (LSCs), followed by an allogeneic donor graft to restore normal hematopoiesis. Additional clearing of LSCs occurs via the graft versus leukemia (GVL) effect of donor stem cells [16]. Mechanisms of relapse commonly involve a pre-transplant dominant clone that is genetically identical or acquires additional driver mutations (Figure 1). In rare cases, the evolution of an undetected pretreatment subclone may cause relapse [17,18,19]. Likewise, the aberrant antigen combinations of LSCs, defined as leukemia-associated immunophenotypes (LAIPs), undergo significant changes at the time of transplantation due to conditioning and the aforementioned GVL effect [20].

It has been suggested that clonal sensitivity to pharmacologic agents leads to the emergence of one or more subclones due to selection pressure induced by HSCT [19]. A study conducted on relapsed MDS patients after HSCT employed deep error sequencing to longitudinal tumor samples and revealed that relapse was characterized by the persistence of dominant clones with expansion or emergence of subclones after HSCT [19]. These emergent subclones are responsible for triggering further clonal evolution at disease progression. Furthermore, it was observed that pretransplant regimens such as hypomethylating agents (HMAs) had an impact on subclonal evolution at progression in MDS [19].

## 3. Statistical Considerations in Interpretation of MRD

MRD test results, although being quantitative, are typically reported with a binary outcome (positive or negative) implying the detection of residual leukemic cells using a minimum threshold depending on the type of assay used [21] (Table 1). An ideal MRD assay requires a sensitivity on the order of 10^−4^ (capable of detecting one abnormal cell in 10,000 cells) to be clinically applicable [22]. An emphasis on the predictive performance of any MRD assay should not ignore these statistical properties. An ideal test should fulfill the benchmark for these without being cumbersome and with potential for wide applicability. In practice, the different MRD assays are considered complementary. Moreover, serial testing decreases the likelihood of false positives and negatives. The ELN guidelines now recommend sequential MRD testing to improve the sensitivity/specificity of the results for eventual clinical relapse [13]. This is due to the fact that a restricted signature of molecular lesions does not take into consideration the heterogeneous behavior of individual AML subclones in response to therapy (either chemotherapy or transplantation) [23].

## 4. Available Strategies for MRD Detection

During the past decade, MRD monitoring has transitioned from an endpoint in clinical trials to being incorporated into routine practice, in part due to modern technologies in monitoring assay methods. Despite technological advances, MRD testing, interpretation, and clinical application are all subject to limitations. The assays for MRD detection rely on either immunophenotyping via multiparametric flow cytometry (MFC) or molecular testing. MFC is the most widely employed method with sensitivity for LAIPs of 10^−4^ to 10^−5^ [28]. A distinct disadvantage of the LAIP-based approach is due to the change in LAIPs over time from either clonal evolution or immunophenotypic changes occurring upon treatment [24].

Current technologies for molecular MRD testing include fluorescence in situ hybridization (FISH), real-time quantitative polymerase chain reaction (RT-qPCR), and NGS. Molecular testing is increasingly being employed to detect leukemia-defining mutations. Among these, FISH and RT-qPCR are well standardized in clinical practice [8]. The use of molecular testing for MRD assessment is now recommended over MFC according to the 2021 ELN MRD guidelines [29]. The heterogeneous nature of AML requires targeted molecular and cytogenetic workup for an accurate diagnosis and prognostication. For instance, *NPM1* and *FLT3* genes are two of the most common targets of genetic mutations found in AML with the former having a relatively favorable prognosis. However, pre-transplant MRD-positive *NPM1*-mutant AML with *FLT3*-ITD (internal tandem duplication) has significantly poorer outcomes [25] such that *FLT3*-ITD is classified as intermediate-risk regardless of the presence of *NPM1* mutation [30]. Methodologies such as PCR and MFC have disadvantages because they may not target all disease-defining changes or have lower sensitivity and specificity than required [13].

Table 1 and Figure 2 summarize the major features among available MRD assays.

## 5. Utility of MRD Testing during HSCT

In the context of transplantation, MRD testing can be performed before HSCT to establish the baseline disease level for improved risk stratification. AlloHSCT is often recommended for intermediate- and high-risk groups in most protocols [13]. MRD-positive patients achieving complete remission have significantly higher rates of relapse than MRD-negative counterparts (82 vs. 30%) [9]. A large cohort metanalysis of 81 publications also demonstrated the prognostic significance of MRD as a surrogate for disease-free survival and OS in AML, as MRD-negative patients were found to have a higher estimated 5 year disease-free survival of 64% compared to MRD-positive patients with a 25% rate. Similarly, those with MRD-negative status had an overall survival rate of 68% compared to 34% in MRD-positive cases [9]. Findings such as these were instrumental for recent ELN MRD guidelines. Indeed, the 2022 ELN guidelines suggest that MRD status, particularly in NPM1-mutant AML, can result in reclassification of patients as intermediate-risk or favorable-risk AML [13].

During the peri-HSCT period, MRD status has been used to determine the intensity of conditioning regimen, as reduced-intensity conditioning (RIC) has been associated with significantly increased risk of relapse compared to a myeloablative conditioning (MAC) regimen in patients with positive NGS-based MRD (NGS-MRD) [11].

After transplantation, MRD assessment is used periodically to primarily monitor disease kinetics/treatment response (Figure 3) [31]. This may be useful to guide maintenance therapy such as venetoclax [32,33]. Notably in FLT3-ITD-mutated AML, improved outcomes with FLT3 inhibitors (e.g., gilteritinib, quizaritinib, and sorafenib) have been observed in individuals who achieve undetectable FLT3 mutant clones while on maintenance therapy [32,33,34,35,36]. MRD assessment could also guide the choice of therapeutic interventions to prevent relapse ([37,38]). Further interventions include donor lymphocyte infusions, HMAs, immunosuppressive agents, intensive chemotherapy, and even a second alloHSCT [39,40,41,42]. HMAs have shown benefit in MRD-positive NPM1-, RUNX1-, and CBF-mutant AML/MDS in delaying hematological relapse [38,43,44,45].

## 6. Next-Generation Sequencing

More recently, NGS has entered the fray with a closer detection rate to PCR tests, which can potentially cover the whole genome for analysis if needed. The feasibility of NGS for detecting MRD in AML was first explored by Thol et al. in 2012 [46]. The inherent power of NGS-MRD is due to the ability for simultaneous multiple-target evaluation of regions often mutated in AML. This allows for better inference of clonal and subclonal MRD burden as opposed to quantitative PCR or MFC, which are limited to only specific clones [10]. Using serial measurements, NGS can also be used to monitor changing mutational dynamics with treatment [10]. The MRD measure from NGS can be expressed in terms of the variant allelic frequency (VAF), measuring the frequency of reads with mutation aligned to a region, and can be highly sensitive depending on the sequencing depth and bioinformatic pipeline used. However, detection MRD of rare subclones is still difficult due to technical challenges [47]. Relatively recently, incorporation of unique molecular identifiers (UMIs) allowed for increased sensitivity in low VAF mutation detection [48,49]. The incorporation of error correction methods that utilize unique molecular indices to discriminate rare variants from protocol artefacts into the NGS platform has also improved the reliability of NGS analysis [47]. The current guidelines do not recommend commercial NGS-MRD in AML in isolation from other methods [13]. However, when used alongside PCR, there are specific instances where NGS assessment of MRD has proven to be particularly useful, supporting inclusion into routine clinical practice [30]. These include *FLT3*-ITD- and *NPM1*-mutated AML with both having been extensively studied for NGS-based MRD assessment [30,50,51].

## 7. Studies Assessing Utility of NGS-Based MRD during Allogeneic HSCT in AML/MDS

Several mostly retrospective studies and a few prospective trials have been published highlighting the relevance of NGS in detecting MRD in AML/MDS and its implication in the peri-transplant period.

### 7.1. Pre-Transplant MRD Monitoring

In pre-transplant NGS-based monitoring, the main issues are the impact of pre-transplant assessment on risk stratification and clinical decision making regarding the use of allogeneic HSCT [10].

In AML, data from a retrospective study by Morita et al. showed that patients with negative flow-MRD could be further risk stratified into distinct prognostic subgroups according to the clearance of somatic mutations at remission [52]. Indeed, superior OS and lower incidence of relapse were observed amongst patients who achieved complete clearance of somatic mutations with intensive induction chemotherapy prior to transplantation. Similarly, it was discerned that MRD^+^ patients had a higher OS post transplantation with no additional benefits conferred to MRD^−^ patients who had HCT transplantation [52]. The study excluded preleukemic variants (discussed later) from the analysis. A similar study by Levis et al. in 2018 observed that the MRD^+^ group had marginally increased OS compared to the MRD^−^ group; however, no appreciable difference was observed between pre- and post-transplant MRD on OS [53]. In an earlier prospective study by Getta et al., no significant difference in OS was observed between MRD^+^ and MRD^−^ groups post allogeneic stem-cell transplantation, but the incidence of progressive relapse among MRD patients was significantly reduced by about 40% [54]. A more recent retrospective study utilizing DNA sequencing of banked pre-transplant AML samples found that persistence of *FLT3-*ITD or *NPM1* variants is associated with a threefold incidence of relapse after first transplant and decreased survival at 3 years [55].

There is an ongoing clinical trial (NCT02756962), the purpose of which is to determine the impact of these persistent somatic mutation detected by NGS in AML patients on clinical decision making in terms of whether to proceed to allogeneic transplant.

In MDS, the application of NGS-based pre-transplant MRD assessment is exploratory at best [47]. However, retrospective studies have demonstrated that the presence of pre-transplant somatic mutations such as TP53, TET2, DNT3A, and IDH2 are associated with inferior survival in MDS [56,57].

### 7.2. Post-Transplant MRD Monitoring

In AML, studies on post-transplant MRD assessment have mostly utilized serial monitoring of samples for this purpose. One such available study demonstrating the utility of NGS-based MRD assessment in a post-HSCT setting was conducted on samples taken at diagnosis, pre-HSCT, and post-HCT on day 21 and at 3, 6, and 12 month intervals [58]. Patients who remained MRD-positive on day 21 post transplantation had significantly worse 3 year OS and increased incidence of relapse [58]. A study by Heuser et al. with a median follow up of 6 years post HSCT observed a 5 year CIR of 53% vs. 26% (*p* < 0.001) for MRD^+^ versus MRD^−^ patients [23].

Other similar studies have demonstrated that known pre-transplant somatic mutations undergo additional changes which can be tracked. For instance, Kohlmann et al. assayed serial samples with RUNX1 mutations using deep sequencing at diagnosis and time of relapse to show that these mutations remained stable over the course of disease, only acquiring novel mutations in other regions of the gene in a minority of patients [59].

A prospective multicenter trial to determine the clinical utility of molecular MRD by testing samples up to 18 months after alloHSCT is now open to accrual (NCT05224661).

In MDS, the impact of post-allogeneic HSCT monitoring of CBL and TET2 genes using amplicon deep sequencing was assessed [60]. The authors showed a higher incidence of relapse among patients with detectable mutations at a median of 6 months after HSCT [60].

## 8. Single-Cell Next-Generation Sequencing

Although bulk sequencing has allowed for omics mapping, resolving intratumoral heterogeneity is nontrivial, since bulk-seq is an aggregate proxy for the individual cell–omics profiles. For this purpose, single-cell sequencing methods have been developed that allow for tagging/barcoding molecular fragments specific to individual cells. However, scSeq is time-consuming, and extracted data can be noisy. Specifically, in the context of DNA sequencing, scDNASeq requires whole-genome amplification from a low DNA amount, which can result in failure to capture individual alleles (allelic dropouts) [61]. Furthermore, mutational artefacts can be introduced during amplification, which needs to be taken into account during variant calling [62]. Nevertheless, scSeq is becoming the gold standard for probing intratumoral heterogeneity in solid tumors and hematologic cancer [63,64,65].

## 9. Clinical Applications of scDNA-Seq in AML/MDS

Single-cell resolution of AML clonality offers several advantages over bulk NGS. In this section, we discuss the mechanisms that allow for more precise characterization of clonal architecture in scDNA-seq.

### 9.1. Identification of Variants Associated with Clonal Hematopoiesis

Single-cell analysis provides for better resolution of age-related clonal hematopoiesis (ARCH) compared to bulk NGS at remission. Dillon et al. first described the application of scDNASeq with simultaneous single-cell antibody–oligonucleotide sequencing to distinguish nonmalignant ARCH from leukemia [66]. This method was applied during remission to delineate rare ARCH variants from clones associated with relapse [15]. ARCH mutations are typically mutations in epigenetic regulators that are observable at a lower VAF than by bulk NGS. These mutations can complicate molecular assessment of MRD and are collectively known as DTA mutations, including *TET2*, *DNMT3A*, *IDH1/2*, and *ASXL1*. Preleukemic ARCH clones are common in AML and increase with age [50]. Successful alloHSCT is expected to eliminate ARCH-related genetic abnormalities, and their persistence after transplantation may indicate the persistence or relapse of the leukemic clone. However, any prognostic significance of DTA mutations in AML is yet to be demonstrated by studies. Heuser et al., using NGS-based MRD monitoring, found DTA mutations that were not eliminated in 17.6% of their patient cohort with no prognostic impact on incidence of relapse or OS [29]. In contrast, MRD with non-DTA mutations was highly predictive of outcomes in a separate analysis. Therefore, ARCH mutations alone may not necessarily predict relapse risk in AML.

Further incorporation of scDNA-seq in peri-transplant MRD detection could significantly improve its predictive power due to the inherent ability to better detect and exclude these rare ARCH variants not associated with relapse.

### 9.2. Phylogenetics

ScDNA-seq allows for better determination of mutation rank in AML compared to bulk-seq methods, resulting in more definitive phylogenetic models. Two studies used high-throughput single-cell proteogenomics on large AML datasets to provide insights into the evolutionary trajectory of AML. Their findings suggest that epigenetic mutations precede mutations in genes related to signaling pathways such as FLT3 and the RAS family. With the notable exception of TET2 mutations, there was very little clonal trajectory when the initial mutation involved genes of the signaling pathway [67]. One probabilistic phylogenetic prediction model known as the single-cell inference of tumor evolution (SCITE) demonstrated both linear and branching models of clonal evolution in AML, with significantly higher clonal diversity occurring among samples with branching clonal evolution [68].

ScDNA-seq can also predict clonal relationships, on the basis of the individual clone composition and frequency of each clone, to deduce that clones of lower frequency evolve from clones with higher frequency [69]. This principle has been leveraged to describe the clonal trajectory of MDS driving its progression to secondary AML [70]. Recent work also correlated mutational frequency with both clinical outcomes and phenotypic differences such as white blood cell counts, LDH levels, and blast abundance in PB [71].

### 9.3. Deconvolution of Mutation Co-Occurrence

The clonal milieu in AML samples often comprises dominant clones that outcompete other minor clones [67], and it has been observed from prior studies using bulk NGS assays that ≥2 allelic variants are associated with reduced leukemia-free survival and OS [72]. Indeed, single-cell analysis of myeloid malignancies has shed more light on which mutational combinations promote clonal expansion. ScDNA-seq can identify the mutational co-occurrences that lead to dominant clones not previously delineated from bulk sequencing.

Miles et al. observed that specific mutational combinations involving NPM1c + FLT3ITD or DNMT3A + IDH2 resulted in clonal dominance, whereas other co-occurring alleles such as NPM1c + RAS did not promote clonal expansion [67]. The number of variants also has an impact on clonal size, as it was observed that samples with double mutant *DNMT3A*/*IDH1* or *IDH2* clones had significantly larger clonal size compared to respective single-mutant clones [67]. Ediriwickrema et al., using targeted scSeq, identified a significantly greater number of co-occurring number of variants at diagnosis in relapsed patients [15]. In addition, the driver mutations in AML were found to be mutually exclusive in occurrence at cellular level [68]. These findings may inform therapeutic strategies during the peri-transplant period to target these clones before they achieve clonal dominance.

### 9.4. Monitoring Clonal Evolution during Treatment

Significant alterations occur in the clonal architecture of AML when the disease becomes resistant to initial treatment. These so-called “clonal sweeps” involve the emergence of a new dominant clone that affects response to further therapy [73]. Prior studies using bulk-seq methods implicated signaling pathway mutations (particularly RAS/RTK/MAPK mutations) as potential mechanisms of resistance [74]. These clonal mechanisms of resistance have also been studied using scDNA-seq methods. In the context of FLT3-mutated AML, for instance, by examining pre- and post-therapy treatment samples of patients with FLT3 mutations on gilteritinib (selective FLT3 kinase inhibitor), an emergent outgrowth of clones with RAS mutations and suppression of clones with FLT3-ITD mutations was observed. This finding has been supported by other studies elucidating the clonal mechanisms of resistance with other chemotherapeutic agents [75,76,77].

### 9.5. Mapping of Genetic–Phenotypic Evolution in AML

In AML, LAIPs on leukemic blasts may serve as therapeutic targets [78]. The prognostic significance of these immunotypes, especially when combined with genotypic analysis, is yet to be explored.

The simultaneous use of scDNA-seq and immunophenotyping (obtained from cell-surface protein expression analysis in mature hematopoietic lineages) can allow for genotype–phenotype correlation among sequenced cells to better dissect intratumor heterogeneity in AML, providing a linkage between phenotypes and the genetic mutations that drive them. Prior studies using single-cell analysis provided significant insight into the impact of genotype on cell-surface expression in leukemia samples. Morita et al. observed a significant correlation between CD34 expression and the mutations present in samples. They found that higher CD34 expression was seen in samples with TP53 mutations, while the opposite was true in samples with NPM1 and IDH1/2 mutations [68].

Demaree et al. applied DAb-seq [79], a tool for joint profiling of DNA and surface proteins in single cells at high throughput. This method yields genotypic information from DNA samples and is distinct from other tools that rely on transcriptomics and antibody sequencing methods, which only capture phenotypic features. When applied in three patients across multiple treatment timepoints, a strong correlation between the AML genotype and corresponding malignant phenotype was observed in one patient.

The potential applications include the capability to be used to identify and track specific immunophenotypic populations before or after HSCT that may be amenable to cell-surface-targeted therapies.

## 10. Shortcomings of Current Techniques of scDNAseq

The major shortcomings of scDNAseq techniques include false variance call rates, allelic drop out (ADO), limited throughput, lack of standardization, and cost.

Both qPCR and MFC-based MRD analyses in AML aim to detect aberrant cell populations at a one in 10,000 frequency [13]. This is particularly important in the post-transplant period, as MRD presence is associated with relapse [80,81]. However, while single-cell-based genomic sequencing techniques can detect cells with genomic aberrations in this frequency range, there are concerns about false variance call rates that may be much higher, which could significantly reduce the utility of this technology for detecting MRD. In particular, Pellegrino et al. showed that the frequency of detected variant alleles that are potential experimental artefacts can be as high as 2.5%, which is considerably higher than the MRD detection threshold [82]. Currently, spiked-in cell lines and multiple control loci with known frequency are used to detect error rates in each experimental design, and the difference in rate between ADO in control lines and samples must be explored further to validate the use of these control methodologies in future experiments [82].

Moreover, the problems created by ADO (referring to an apparent loss in heterozygosity when an allele at a particular genetic locus fails to amplify or sequence) and false variance calls in general are exacerbated by complex karyotypes and unbalanced chromosomal abnormalities observed in AML genomic landscape, which may cause false variance call rates to be significantly different than the rates observed in commonly used control cell lines and between patient samples [83,84]. Therefore, the optimization of control methods and the demonstration of whether they can be carried over to the complex landscape of genomics in AML clinics must be shown and approved by a consensus of physicians to harmonize use of this technology as a diagnostic tool in MRD.

Inference of clonal architecture through scDNAseq produces better results compared to bulk NGS. However, false-negative variant calls can be significantly reduced by comparing the frequency of cells with variant alleles to VAF of the same mutations in bulk sequencing in the same population [82]. This approach is more reliable in detecting low-frequency MRD, with general error rates for conventional NGS around 0.1% [85,86]. Although this rate causes NGS to fall behind other PCR techniques in the sensitivity of detection of low VAF mutations, it is comparable with MCF [13]. Therefore, bulk DNA sequencing with NGS makes an ideal companion to scDNA-seq for MRD studies. The efficacy of this approach was shown with a novel in silico method called B-SCITE [87], which combined genomic data from the whole cell population using bulk sequencing and single-cell resolution data generated through single-cell genomic sequencing. The authors demonstrated that their model outcompetes multiple scDNA-seq analysis approaches in inferring clonal architecture [87]. Despite the aforementioned shortcomings of each approach, this study showed that there is a measurable benefit to combining these methods, which cannot be achieved through the application of only one.

Another shortcoming of sc-DNAseq for the detection of low-frequency MRD is the necessity for ultrahigh-throughput sequencing. For a one in 10^4^ detection rate, upward of 10,000 cells must be sequenced, ideally many times more to make up for losses throughout the experimental workflow. sc-DNAseq experiments of myeloid malignancies have not incorporated as many cells in experiments [15,67,68,82]. However, the robustness of MRD detection requires high-throughput sequencing, and the establishment of clonal hierarchy and phylogeny in tumor subclones does not. The frequency of studied mutations is significantly higher than the lower limit of detection of MRD. Nonetheless, established workflows can reach necessary cell read counts [88], but the benefits of using such ultrahigh-throughput sequencing in the setting of hematological malignancies must be established before the high cost of usage can be justified. The primary utility of scDNAseq in MRD detection will not be detection of ultralow-frequency residual disease. As a result, a combination of bulk and scDNA sequencing can mitigate the problem of detection of low-frequency variants through carefully prepared high-depth NGS panels [89].

The need for a good cellularity sample may pose a challenge due to hypocellularity occurring during the conditioning/recovery phase following transplantation. This is a peculiar shortcoming of single-cell studies in MDS research in cases of hypocellular MDS. This leads to a paucity of cells that will create a major challenge for all single-cell platforms, along with the heterogeneity of disease, with multiple clones carrying different sets of somatic mutations in each patient.

Another hurdle to overcome would be to reduce the heterogeneity of sizes and ranges of the currently available panels, which range between 19 and 685 genes being used for variant allele detection in scDNAseq studies in AML so far [67,68,76,82]. If scDNAseq is to be exported to the clinical realm of MRD detection in AML, commercially available panels should include at least the most up-to-date somatic mutations that are defined by ELN, at a sensitivity that is comparable with established methods for MRD detection [13]. If clonal hierarchy can be established with the help of single-cell genomics while keeping the MRD detection rate stable through integrative genomics with bulk NGS, further research will need to establish how the information gathered through deciphering clonal hierarchy in each patient can help best utilize precision treatment options.

Lastly, we acknowledge the cost of scDNAseq as a barrier to the expansion of AML MRD detection for clinical use. scDNAseq is expensive, including procedural costs such as sample collection and library preparation, as well as the analysis and interpretation of the output requiring skilled personnel [90]. A lack of standardization in mutation panels adds an additional level of complication. Low insurance coverage and reimbursement rates will further hinder scDNAseq utilization, as insurance companies will still consider the application of scDNAseq MRD testing experimental. Targeted sequencing of frequent AML mutations by encapsulation and barcoding of single cells is expected to decrease the cost threshold of scDNAseq in the near future [90].

## 11. Single-Cell RNA Sequencing (ScRNAseq)

Throughout this review, we tried to uncover the potential of scDNA-seq in materializing the clonal architecture and hypothesize the benefits that can be observed in the clinical setting. However, scRNA-seq is an additional single-cell based analysis method that can shed even more light on the disease state beyond genomic alterations. Such transcriptomic analyses have been utilized in solid tumors [91] and are being explored in clinical settings for hematologic malignancies [92,93,94]. So far, studies in AML have successfully identified similar cell-types in different patients showing that, even with significant intratumoral heterogeneity [95], there are co-upregulated pathways in different cell subtypes in different patients that behave similarly, and the transcriptomic patterns of these different cell types correlate with underlying genetic alterations. The effect of genomic aberrations over time, especially crucial ones such as *FLT3*, *IDH1*, *IDH2*, and *NPM1*, on the transcriptomic profile of different cell subtypes can be analyzed using a combination of transcriptomic and genomic analysis, which can improve our understanding of pathways involved in relapse, development of resistance, and response to therapy. Targetable transcriptomic changes in crucial genes that regulate proliferation and survival in AML/MDS can help generate more personalized therapy and monitor response to treatment in both clonal cells and healthy immune cells prior to relapse at a much higher resolution that is obtainable by bulk sequencing methods [96,97] (Figure 4).

scRNAseq can observe the effect of mutations, such as *RUNX1*, *NPM1*, *CBFB*, and other genetic abnormalities, on therapy response. Differences in transcriptomic profiles of cellular subgroups with distinct expression profiles can be analyzed to observe the effect of targeting these mutations on the transcriptomic landscape. During the post-transplant period, activation of proliferative transcriptomic signatures in healthy immune cells may indicate a favorable prognosis; however, in cell populations carrying the aforementioned mutations, it could be an early sign of relapse.

In MDS, studies utilizing scRNAseq have generated transcriptomic profiles of hematopoietic stem cells and multipotent progenitor cells from MDS patients and identified differentially expressed genes regulating myeloid differentiation patterns, RNA metabolism, and ribosome biogenesis compared to healthy donors [98]. Understanding the transcriptomic landscape can provide insights into the progression patterns of cells with somatic mutations in MDS. This information can aid in the integration of personalized therapy options or the development of diagnostic panels for patients with MDS or secondary AML.

## 12. Single-Cell Analysis of Accessible Chromatin (scATACseq)

Single-cell analysis of accessible chromatin (scATACseq) has several potential applications in AML/MDS.

Chromatin and gene expression signatures can provide prognostic information about disease progression in hematologic cells at a single-cell resolution. A combinatorial approach using ATAC-seq and RNA-seq at the single-cell level can identify epigenetic regulatory changes. This approach revealed the conservation of transcription factors such as RUNX1, even in phenotypically heterogeneous cells in each patient from one study [99], demonstrating that, in mixed-phenotype acute leukemia (MPAL), the entire mutational landscape can arise from a single multipotent progenitor cell. In pediatric AML, scRNAseq and ATAC-seq revealed the priming of high-risk proliferation pathway genes, even without significant changes in transcription levels, suggesting potential investigation of MRD at earlier disease stages [100]. These sequencing technologies can also analyze nonclonal immune cells, providing accurate mapping of transcriptomic and epigenomic changes that contribute to AML progression and therapy resistance [101].

The chromatin accessibility landscape can be used to identify preleukemic HSCs, LSCs, and leukemic blast cells. By combining RNA-seq and ATAC-seq, these unique cell populations can be distinguished from clonal heterogeneity and observed in an intermediate state of development, in a regulatory state not seen in healthy myelopoiesis [102]. The combinatorial approach helps identify unique cell populations in AML that cannot be explained through intercellular heterogeneity in clonal progression.

Lastly, scATACseq is a useful tool for capturing the regulatory profile of clones in MDS, as previously described in this review [103]. In addition, using single-cell technologies to describe gene expression and regulation in clones with multiple mutations, such as RUNX1 and ASXL1, can help elucidate the functional aberrations caused by these mutations that are associated with the clinical findings in MDS (namely, cytopenias) [104].

## 13. Conclusions

Molecular assessment of MRD during the peri-transplant period using NGS-based technologies allows for early detection of disease recurrence, which can help guide treatment decisions and potentially improve patient outcomes in AML/MDS. scNGS could emerge as a powerful prognostic tool that complements existing assays for MRD detection, leading to better risk stratification and proactive treatment of resistant clones.

## Figures and Tables

**Figure 1 cancers-15-02477-f001:**
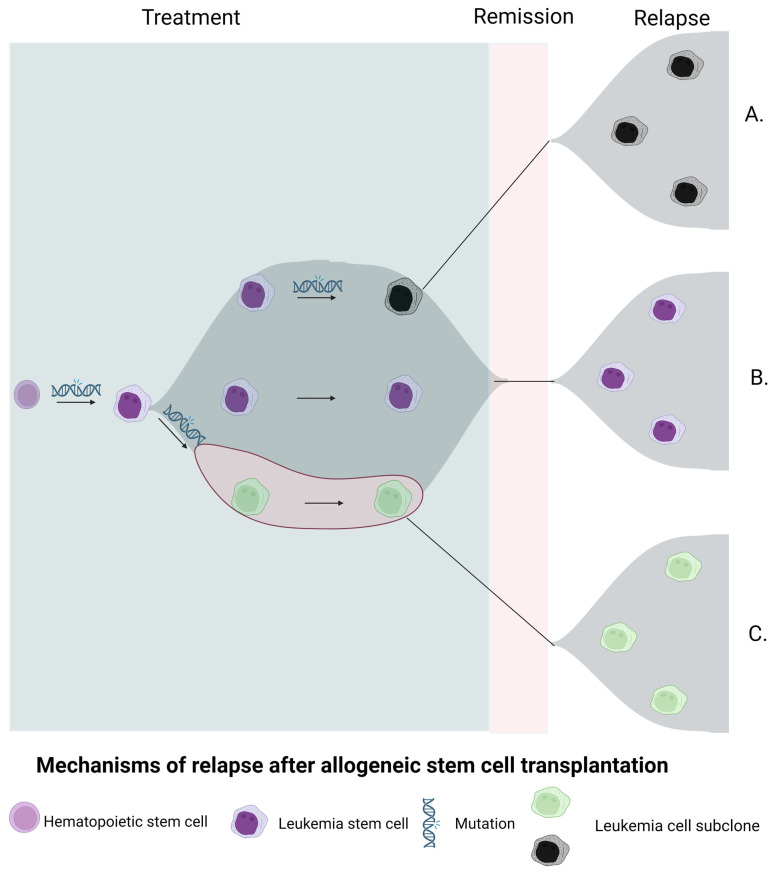
Mechanisms of relapse in AML/MDS after allogeneic stem-cell transplantation. (**A**) Relapse involving a pre-transplant minor subclone not detected by conventional sequencing techniques. (**B**) Relapse involving a pre-transplant dominant clone. (**C**) Relapse involving a pre-transplant dominant clone acquiring additional driver mutations detected and tracked through transplant process.

**Figure 2 cancers-15-02477-f002:**
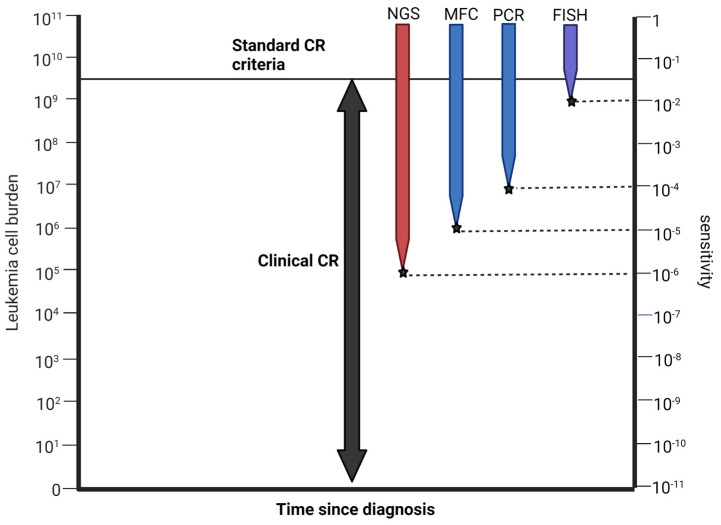
Sensitivities of various MRD detection methodologies.

**Figure 3 cancers-15-02477-f003:**
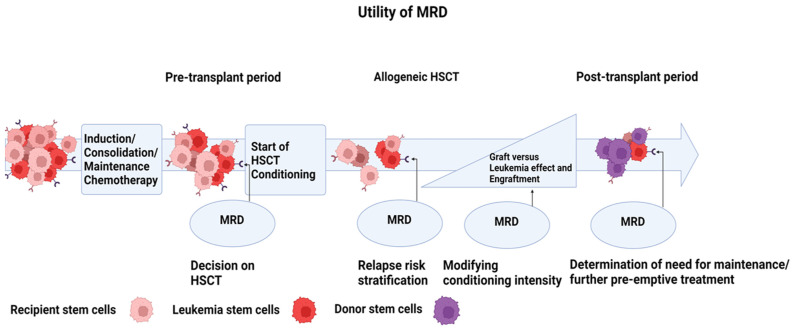
Utility of MRD assessment during the transplant period.

**Figure 4 cancers-15-02477-f004:**
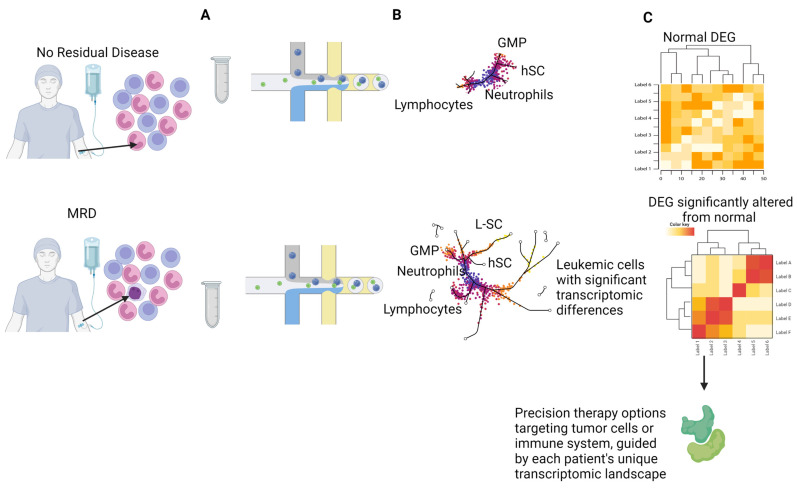
MRD scRNA diagnostic advantage. DEG: differentially expressed genes, GMP: granulocyte monocyte progenitor, HSC: hematopoietic stem cell, L-SC: leukemia stem cell, MRD: minimal residual disease. (**A**) Microfluidics system barcoding cells. (**B**) Upper panel, patient without MRD with transcriptome of regular peripheral blood; no clonal lines detected. Bottom panel, clustering of cell lines from peripheral blood of patient with MRD through dimensional reduction of high-dimensional transcriptional data; clusters include pathological clonal lines including L-SC. (**C**) Panel colors represent gene expression. Upper panel, no differential expression of genes. Bottom panel, differentially expressed genes in MRD. Significantly elevated expression of targetable genes might increase the benefit derived from precision medications.

**Table 1 cancers-15-02477-t001:** Features and merits of MRD detection methodologies.

Method	Target and Clonal Consideration	Applicable in % of AML	Turn-Around Time (Days)	Advantages	Technical Limitations
Multiparameter flow cytometry (MFC)	Leukemia-associated immunophenotype (LAIP) Considers all clones with identical phenotype	Nearly 100%	2	Rapid turnaround time and wide applicability	Less sensitive and more subjective analysis LAIPs change over time [24]
Real-time quantitative PCR (RT-qPCR)	Specific targeted mutations Validated molecular targets: *NPM1*, *CBFB*::*MYH11*, *RUNX1*::*RUNX1T1* Less robust data: *KMT2A*::*MLLT3*, *DEK*::*NUP214*, *BCR*::*ABL1*, *WT1* [9,25,26,27] Detects only a single clone	60–70%	3–5	Comparable sequential results Widely accepted Risk stratification in specific mutant types to determine relapse risk, e.g., NPM1-mutant AML [25]	Specific assay necessary for every mutation Subject to amplification bias Restricted molecular targets
Next-generation sequencing (NGS)	Potentially any somatic mutation Considers all minor clones with infrequent occurrence	≥90%	5–10	MRD monitoring during allogeneic HSCT	Expensive Technically challenging Complex bioinformatics involving several steps such as quality control of raw data, library preparation, and variant calling

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
