# Peer review of "Single-Cell Next-Generation Sequencing to Monitor Hematopoietic Stem-Cell Transplantation: Current Applications and Future Perspectives"

_cancers, 2023, doi:10.3390/cancers15092477_

Round 1

Reviewer 1 Report

NA

NA

Author Response

We are grateful for reviewer's insightful comments

Reviewer 2 Report

Ogbue et. al. comprehensively describes the challenges and potentials of using the Single cell sequencing approaches upon treatment of leukemia. The review particularly focuses on measurable residual disease (MRD) assessement in AML and MDS after hematopoietic stem cell transplantion. It provides a nice overview about the currently applied techniques and emphasizes/promotes the usage of single cell NGS for monitoring genetic changes during the disease. The review is well written and easy to follow. The figures are helpful.

Some minor points:

1. The review does not address the costs and necessary equipment for the various techniques. Perhaps a short discussion how realistic it is to apply these techniques in the clinic may be useful. 

2. In table 1 it is mentioned that the NGS technique requires "complex bioinformatics". Perhaps this can be further elaborated.

3. I would recommend to shorten the title. Perhaps: "Single-Cell Next Generation Sequencing to Monitor Hematopoietic Stem Cell Transplantation: Current and Future Perspectives". 

4. Figure 3 and 4 should/can use the entire page size. 

5. The figure legends are in my view too brief (especially if Figure 3 and 4). An expansion to explain better what is shown may be helpful.

6. Typo: Abstract: "not currently used r in the clinical setting." > " not currently used in the clinical setting."

Author Response

Reviewer 2:

  1. The review does not address the costs and necessary equipment for the various techniques. Perhaps a short discussion how realistic it is to apply these techniques in the clinic may be useful.

Response:

We thank the reviewer for this comment. We have added a section addressing the cost prohibitive nature of scDNAseq.

Page 14: Line 481-488

“Finally, we acknowledge the cost of scDNAseq being a barrier for the expansion of AML MRD detection for clinical use. scDNAseq is expensive, including procedural costs such as sample collection and library preparation, as well as the analysis and interpretation of the output requir skilled personnel (PMID 31391919). A lack of standardization in mutation panels adds an additional level of complication. Low insurance coverage and reimbursement rates will further hinder scDNAseq utilization as insurance companies will still consider the application of scDNAseq MRD testing.  Targeted sequencing of frequent AML mutations by encapsulation and barcoding of single cells are expected to decrease the cost threshold of scDNAseq in the near future (PMID 31391919).”

Page 12: Line 427-434

  1. In table 1 it is mentioned that the NGS technique requires "complex bioinformatics". Perhaps this can be further elaborated.

Response:

We thank the reviewer for this relevant comment. We elaborated further in Table 1 (limitations for NGS):

“Complex bioinformatics involving several steps such as quality control of raw data, library preparation and variant calling.”

Table 1

  1. I would recommend to shorten the title. Perhaps: "Single-Cell Next Generation Sequencing to Monitor Hematopoietic Stem Cell Transplantation: Current and Future Perspectives".

We thank the reviewer for this important comment. The title has been modified accordingly.

“Single-cell Next Generation Sequencing to Monitor Hematopoietic Stem Cell Transplantation: Current Applications and Future Perspectives”

Page 1 Line 1

  1. Figure 3 and 4 should/can use the entire page size.

Response

We thank the reviewer for this valuable comment. Figures 3 and 4 have been modified accordingly.

  1. The figure legends are in my view too brief (especially if Figure 3 and 4). An expansion to explain better what is shown may be helpful.

Response

We thank the reviewer for pointing this out. We have modified the figure legends accordingly:

“Figure 3. Utility of MRD assessment during the transplant period. Pre-transplant period: Decision on HSCT and relapse risk stratification; Peri transplantation: Modifying conditioning regimen; post-transplant period: Determination of need for maintenance/further treatment.

Figure 4. MRD scRNA diagnostic advantage. DEG: Differentially expressed genes, GMP: Granulocyte Monocyte Progenitor, HSC: Hematopoietic Stem Cell, LSC: Leukemia stem cell, MRD: Minimal Residual Disease.

  1. Typo: Abstract: "not currently used r in the clinical setting." > " not currently used in the clinical setting."

Response

We are grateful to the reviewer for noting this. The typographical error was corrected.

Page 1, Line 36

Reviewer 3 Report

The manuscript reviews MRD application in AML and MDS, particularly with high sensitivity NGS tools; it is generally well written and readable, with technical insight about novel assays and combination of different techniques. There are a few typos to correct and in my opinion the figures could be improved. 

In particular:

abstract, row 37: please correct the typo (an extra "r")

page 2 row 81: sentence not clear

please harmonize “allogenic”/“allogeneic” in the text

figure 1 requires some changes as it is not entirely clear (in particular the emergence of new mutations in option C needs to be highlitghted)

page 4, row 119: please change “subject of” to “subject to”

page 4 rows 122-123: it could probably be added that another disadvantage of may MRD tools is the need for a good cellularity sample, which might be difficult to obtain particularly during hematologic recovery after treatment, including transplantation

page 5 row 144: please correct the sentence

page 5 row 162: when discussing maintenance, different options than venetoclax should be cited, particularly for FLT3+ patients

figure 3 is also not so clear and should be modified, highlighting the pre-transplant sample as an informative MRD timepoint as well; visual aspects are not well addressed in my opinion

page 7 row 203 sentence not clear

pare 8 rows 286-290: sentence not clear

page 9, row 311: please correct the sentence

page 9, row 343: please correct the sentence

page 9 rows 345-347: I wouldn’t emphasize such preliminar results on only one patient

figure 4: abbreviations should be defined, and, possibly figure should be clarified

Author Response

Reviewer 3:

abstract, row 37: please correct the typo (an extra "r")

Response

We are grateful to the reviewer for noting this. The typographical error was corrected at:

Page 1, Line 36

page 2 row 81: sentence not clear

Response

We thank the reviewer for this observation. We have modified the sentence

“Mechanisms of relapse commonly involve a pre-transplant dominant clone that is genetically identical or acquires additional driver mutations (Fig.1). In rare cases, the evolution of an undetected pre-treatment subclone may cause relapse.”

please harmonize “allogenic”/“allogeneic” in the text

Response

We thank the reviewer for this comment. We have harmonized the text and figures to “allogeneic”

figure 1 requires some changes as it is not entirely clear (in particular the emergence of new mutations in option C needs to be highlitghted)

Response

We thank the reviewer for this observation, Figure 1 has been modified to highlight the emergence of subclones.

page 4, row 119: please change “subject of” to “subject to”

Response

We thank the reviewer for this observation. Sentence has been modified.

Page 4 line 121

page 4 rows 122-123: it could probably be added that another disadvantage of may MRD tools is the need for a good cellularity sample, which might be difficult to obtain particularly during hematologic recovery after treatment, including transplantation

Response

We thank the reviewer for this helpful comment. We have incorporated this comment in the section:

“The need for a good cellularity sample may pose a challenge due to hypocellularity occurring during conditioning/recovery phase following transplantation. This is a peculiar shortcoming of single cell studies in MDS research in cases of hypocellular MDS.”

Page 12 line 413-418

page 5 row 144: please correct the sentence

Response

The statement and figure legend was modified.

Page 5 line 144.

page 5 row 162: when discussing maintenance, different options than venetoclax should be cited, particularly for FLT3+ patients

Response

We thank the reviewer for this insightful comment. We have cited other agents used for maintenance therapy in FLT3 mutated AML:

“Notably in FLT3-ITD mutated AML, improved outcomes with FLT3 inhibitors e.g., gilteritinib, quizaritinib and sorafenib have been observed in individuals who achieve undetectable FLT3 mutant clones while on maintenance therapy.”

figure 3 is also not so clear and should be modified, highlighting the pre-transplant sample as an informative MRD timepoint as well; visual aspects are not well addressed in my opinion

Response

We thank the reviewer for this important comment. We modified figure 3 to highlight the utility of MRD within the time point of interest namely- Pre-transplant, peri-HSCT and post-transplant.

See Figure 3

page 7 row 203 sentence not clear

Response

This sentence was modified.

“In pre-transplant NGS-based monitoring, the main issues are the impact of pre-transplant assessment on risk stratification and clinical decision-making regarding the use of allogeneic HSCT.10

Page 8 Line 206-208

pare 8 rows 286-290: sentence not clear

Response

We adjusted this sentence in response to the reviewers’ insightful comments.

“ScDNA-seq allows for better determination of mutation rank in AML compared to bulk-seq methods, resulting in more definitive phylogenetic models. Two studies used high-throughput single-cell proteogenomics on large AML datasets to provide insights into the evolutionary trajectory of AML. Their findings suggest that epigenetic mutations precede mutations in genes related to signaling pathways such as FLT3 and the RAS family.”

Page 9 Line 291-296

page 9, row 311: please correct the sentence

Response

This sentence was corrected in response to reviewer’s observation:

“Miles et al. observed that specific mutational combinations involving NPM1c + FLT3ITD or DNMT3A + IDH2 resulted in clonal dominance, whereas other co-occurring alleles such as NPM1c + RAS did not promote clonal expansion.63

Page 10 Line 316-318

page 9, row 343: please correct the sentence

Response

We thank the reviewer for this helpful comment. The sentence was modified:

“Morita et al observed a significant correlation between CD34 expression and the muta-tions present in samples. They found that higher CD34 expression was seen in samples with TP53 mutations, while the opposite was true in samples with NPM1 and IDH1/2 mutations.”

Page 11 Line 347-350

page 9 rows 345-347: I wouldn’t emphasize such preliminar results on only one patient

Response

We agree with the reviewer regarding this statement. The statement in question has been removed.

figure 4: abbreviations should be defined, and, possibly figure should be clarified

Response

Figure 4 was modified accordingly.